# DUX4 Role in Normal Physiology and in FSHD Muscular Dystrophy

**DOI:** 10.3390/cells10123322

**Published:** 2021-11-26

**Authors:** Emanuele Mocciaro, Valeria Runfola, Paola Ghezzi, Maria Pannese, Davide Gabellini

**Affiliations:** Gene Expression and Muscular Dystrophy Unit, Division of Genetics and Cell Biology, IRCCS San Raffaele Scientific Institute, 20132 Milano, Italy; mocciaro.emanuele@hsr.it (E.M.); runfola.valeria@hsr.it (V.R.); ghezzi.paola@hsr.it (P.G.); pannese.maria@hsr.it (M.P.)

**Keywords:** FSHD, DUX4, ZGA, B-ALL, DUX4-IGH, CIC-DUX4

## Abstract

In the last decade, the sequence-specific transcription factor double homeobox 4 (DUX4) has gone from being an obscure entity to being a key factor in important physiological and pathological processes. We now know that expression of DUX4 is highly regulated and restricted to the early steps of embryonic development, where DUX4 is involved in transcriptional activation of the zygotic genome. While DUX4 is epigenetically silenced in most somatic tissues of healthy humans, its aberrant reactivation is associated with several diseases, including cancer, viral infection and facioscapulohumeral muscular dystrophy (FSHD). DUX4 is also translocated, giving rise to chimeric oncogenic proteins at the basis of sarcoma and leukemia forms. Hence, understanding how DUX4 is regulated and performs its activity could provide relevant information, not only to further our knowledge of human embryonic development regulation, but also to develop therapeutic approaches for the diseases associated with DUX4. Here, we summarize current knowledge on the cellular and molecular processes regulated by DUX4 with a special emphasis on FSHD muscular dystrophy.

## 1. Introduction

*Double homeobox 4* (DUX4) encodes for a transcription factor with increasingly important roles in normal physiology and in disease.

During the early steps of embryogenesis, DUX4 is involved in the regulation of genes important for pre- and post-implantation development [1,2,3]. Subsequently, DUX4 expression is silenced in most adult tissues, with the exception of testis and thymus [4,5], where it performs unknown functions.

DUX4 aberrant reactivation or gene rearrangements are associated with several diseases, including infection by viruses of the Herpesviridae family [6,7], acute lymphoblastic leukemia [8,9,10], undifferentiated round cell sarcoma [11], several neoplasms [12,13,14] and facioscapulohumeral muscular dystrophy (FSHD) (Figure 1) [15].

Given its importance in physiological and pathological states, there is great interest in understanding how DUX4 expression and activity are regulated. This is particularly relevant for FSHD because the disease displays an enormous variability in age of onset, rate of progression and severity, even among family relatives [16].

Here, we summarize the biological role of DUX4 and its involvement in FSHD with an emphasis on the cellular and molecular aspects that contribute to the disease and could be targeted for therapeutic purposes.

## 2. Physiological Role of DUX4

### 2.1. DUX4 Protein Structure and DNA Binding

Double homeobox 4 (DUX4) is a transcription factor belonging to a larger family (DUXA, DUXB, DUXC, Dux and Duxbl) exclusive to mammals and encoded by repetitive elements [17,18]. DUX4 presents two closely spaced 60-aminoacid homeodomains, HD1 and HD2, which are located at its N-terminus and responsible for sequence-specific DNA binding (Figure 2A). The homeodomain is an ancient DNA binding motif found in animals, plants and fungi, and is present in many transcription factors, including the Hox transcription factors responsible for the generation of morphological diversity along the anteroposterior body axis during development [19]. The origin of double homeodomain proteins is probably due to a gene duplication within a single homeobox common progenitor [18]. Accordingly, in DUX4, HD1 and HD2 share a high sequence similarity [20,21].

The 3D structure of DUX4, predicted using AlphaFold tool [22], shows that the two homeodomains and the C-terminal region, containing the DUX4 transcription activation domain, have a well-defined tertiary structure. Instead, the region between HD2 and the C-terminus is predicted to be disordered (Figure 2B).

DUX4 presents at least three nuclear localization signals (NLS) located in the homeodomains region and an additional one in the last 100 amino acids at the C-terminus, redundantly working to assure nuclear entrance [23].

Since DUX4 binding to DNA is crucial for its role both in physiology and in disease, structural studies elucidating how DUX4 binds to DNA and regulates its targets are extremely important, and could lead to the development of therapeutic strategies, such as small molecules modulating DUX4 interaction with DNA [24]. Crystallography studies have analyzed in detail the structure of the two DUX4 homeodomains in complex with DNA. DNA binding by DUX4 occurs in a head-to-head manner symmetrically over the DNA axis. Despite their high sequence and structure similarity, HD1 and HD2 recognize different core sequences: 5′-TAAT-3′ and 5′-TGAT-3′ in the consensus 5′-TAATCTAATCA-3′, respectively [21]. HD1 and HD2 present a canonical three α-helical structure, typical of homeodomains. The third helix (α3), notably longer in HD1, recognizes the major groove of DNA making direct hydrogen-bond interactions. Instead, the interaction with the minor groove is promoted by the N-terminal loop that precedes the first helix (α1) both in HD1 and HD2 (Figure 2B). Moreover, the interaction between the homeodomains and the DNA is coordinated and stabilized by the linker connecting HD1 and HD2. Additionally, the interaction is promoted by the presence of positively charged residues at DUX4 N-terminus that support the binding to DNA that exhibits an intrinsic negative charge. DUX4 binding to its consensus target region, which is mediated by the folding of the homeodomains covering three consecutive grooves and embracing the DNA, had never been observed before in other transcription factors with α-helical binding domains [20,21,25].

### 2.2. DUX4 Activity in Embryo

After fertilization, the zygote undergoes cell cycles and cell divisions with no significant overall growth (cleavage stage). The initial steps of the cleavage stage take place without gene transcription due to maternal RNAs and proteins contributed by the oocyte. Subsequently, the maternal material is degraded and development continues owing to the first major wave of embryonic gene transcription called zygotic genome activation (ZGA) [26]. Intriguingly, DUX4 accumulates after fertilization showing a peak at the 4-cell stage and being strongly downregulated at RNA and protein level at the 8-cell stage [1,3]. RNA sequencing (RNA-seq) and chromatin immunoprecipitation sequencing (ChIP-seq) demonstrated that DUX4 directly activates the transcription of several protein-coding genes transiently expressed in cleavage stage embryos, including *ZSCAN4*, *KDM4E*, *PRAMEF* and *ZFP352* [1]. DUX4 also drives expression of endogenous retroviruses, such as *HERVL*, which are known to be selectively transcribed at the cleavage stage [27]. Similar results were obtained for DUX4 functional homolog in mouse, Dux [3]. This is intriguing considering that Dux and DUX4 HDs are not highly conserved and that mouse and human *ERVL* sequences are different. Indeed, when expressed in mouse, DUX4 fails to activate ZGA-related *ERVL*. It is tempting to speculate that the function of the ancestral DUX protein was to regulate embryonic gene transcription, and Dux and DUX4 acquired the ability to regulate *ERVL* coevolving with their respective species-specific transposon targets through convergent evolution [2].

While Dux/DUX4 ectopic expression in mouse/human embryonic stem cells is sufficient to drive the expression of ZGA genes, not all mouse/human ZGA genes become activated upon Dux/DUX4 expression [1,2,3,28]. Hence, while Dux and DUX4 are important ZGA regulators, they are not the only players. Accordingly, homozygous *Dux* knockout mice display pre- and post-implantation development defects and significant rates of late embryonic mortality. However, viable litters from these mice can occur, indicating that Dux is involved but not strictly essential for early embryo development [29,30,31,32].

Recently, a role for Dux/DUX4 in the maintenance of genome stability in early embryos has been reported [33,34]. Accordingly, Dux expression is tightly controlled at DNA, RNA and protein levels and its prolonged activation leads to developmental arrest and embryo death [30,35].

### 2.3. DUX4 in Somatic Tissues

Understanding the physiological role of DUX4 and its regulation in somatic tissues is relevant to develop safe and effective treatments for DUX4-associated diseases.

While DUX4 is silenced in most adult tissues, its transcript and protein have been found at relatively high levels in human testis and thymus [4,5]. Although the exact function of DUX4 in these contexts is not known, a common aspect to testis and thymus is the high apoptosis rate characterizing developmental processes undergoing in these two tissues. Spermatogenesis is a complex process requiring homeostasis of different cell types. Accordingly, 75% of germ cells undergoes apoptosis at various stages of testis development and sperm differentiation [36,37]. In thymus, successful rearrangement of the T cell receptor β-chain is required for proper T cell development and cells with a non-functional rearrangement undergo apoptosis. Intriguingly, Duxbl, another DUX4 homolog in mouse, is selectively expressed just before β-selection and plays an important role in elimination of cells that have failed rearrangement of the T cell receptor β-chain [38,39].

Testis and thymus are both associated with a defined immune status; indeed testis is an immunologically privileged site and thymus is involved in the production and maturation of immune cells. However, whether DUX4 is involved in these processes is currently unknown and deserves future studies. 

Several studies have combined ectopic DUX4 expression with transcriptomics and (in some cases) ChIP-seq to identify DUX4 target genes [40,41,42,43,44,45,46,47,48,49]. Surprisingly, the overlap between transcriptomic and ChIP-seq datasets is relatively low, possibly due to different experimental conditions (discussed in [50]).

Gene ontology analyses of the transcripts modulated by DUX4 across different studies highlight several processes including cell differentiation, proliferation, RNA transcription, RNA processing, cytoskeleton organization, immune response and viral response [43,49] (Figure 3). Overall, the DUX4-associated transcriptional signature is in line with its role in establishing an early embryonic program by the negative regulation of cell differentiation and the positive regulation of cell proliferation. It is conceivable that the embryonic program induced by DUX4 is incompatible with skeletal muscle differentiation and might contribute to the increased apoptosis triggered by DUX4 aberrant expression in FSHD leading to muscle degeneration (see Section 4).

Direct DUX4 targets include several transcription factors, such as DUXA, DUXB, LEUTX or ZSCAN4, and other regulators of gene expression, including KDM4E, MBD3L2, MBD3L3 or MBD3L5. Intriguingly, some of these factors can regulate each other, potentially amplifying the transcriptional cascade induced by DUX4 [51,52]. Among DUX4 targets are also the histone variants H3.X and H3.Y, which subsequently incorporate into the genomic regions of DUX4 target genes, thus facilitating the expression of DUX4 targets subsequent to a brief pulse of DUX4 expression [53]. Hence, even a relatively low and transient expression of DUX4 could establish a memory and an amplification of its transcriptional network. The possibility that DUX4 downstream pathways might persist in the absence of DUX4 raises concern regarding the efficacy of therapeutic approaches aimed at targeting DUX4 expression or activity (see Section 10).

Beside protein-coding genes, DUX4 also activates the expression of several classes of repetitive elements including *Alu*, *LINE-1*, *mammalian apparent LTR-retrotransposons* (*MaLR*), *endogenous retrovirus elements* (*ERVs*) and *pericentric human satellite II* (*HSATII*) repeats [1,2,40,41,54]. Some DUX4-activated *MaLR* and *ERV* elements give rise to retrotransposon transcripts, novel promoters for human protein-coding genes, long non-coding RNAs or antisense transcripts [40,41]. While these DUX4-activated transcripts are present and required for embryonic development, their aberrant expression in muscle cells could contribute to FSHD pathology (see Section 4).

## 3. Pathological Role of DUX4

DUX4 expression is regulated by several factors [55,56] and its post-natal aberrant reactivation and/or activity has been associated with several disease states [6,8,11,12,15].

Historically, the first disease that has been associated with DUX4 is FSHD (MIM: 158900, 158901), one of the most prevalent neuromuscular disorders [57]. The disease is characterized by progressive skeletal muscle weakness and wasting, but displays high intra- and interfamilial variability in age of onset, symptoms, presentation and progression [58,59,60,61,62].

FSHD is associated with reduction in copy number and/or loss of epigenetic silencing of the *D4Z4* repeat array located in 4q35. In healthy subjects, the *D4Z4* array displays high copy number and the FSHD locus is epigenetically silenced. In ~95% of FSHD patients, the disease is associated with deletions leaving 1–10 residual *D4Z4* units on one chromosome 4 (FSHD1, MIM: 158900). In the remaining patients, there is no *D4Z4* deletion and the disease segregates with mutations in genes encoding for some of the factors involved in the epigenetic repression of the FSHD locus (*DNMT3B*, *LRIF1* and *SMCHD1*) (FSHD2, MIM: 158901). The two genetic forms of FSHD display loss of epigenetic silencing leading to the aberrant reactivation of the *DUX4* gene, and share common clinical features. Notably, two major 4q subtelomeric allelic variants exist (4qA and 4qB), but only the 4qA allele provides a polymorphic *DUX4* polyadenylation signal, which makes it permissive for DUX4 expression in skeletal muscle [63,64]. As described below, in FSHD, DUX4 has been associated with activation of pathways toxic to muscle tissue [50,65,66], including oxidative stress and DNA damage [67,68], inhibition of myogenic differentiation [67,69,70,71], impaired transcript quality control [54,72,73] and inflammation [40], leading to muscle cell apoptosis [67,74,75].

Aberrant *DUX4* reactivation has been recently described in several forms of solid cancer [12]. In this context, DUX4 drives an aberrant program of embryonic gene expression and provides a distinct advantage to cancer cells, allowing suppression of MHC class I-dependent antigen presentation, immune evasion, and resistance to immune checkpoint blockade [13]. In *DUX4*-positive cancers, a reduced immune infiltration is in contrast with the frequent presence of inflammation and lymphocytic infiltration observed in FSHD in which the inflammatory response has been directly associated with DUX4 expression [40,76,77]. Ectopic expression of DUX4 in human myocytes activates immune-related genes [40], and DUX4 target genes are upregulated in actively inflamed FSHD muscle biopsies, identified by magnetic resonance imaging (MRI) as short tau inversion recovery (STIR) positive [78,79]. Inflammation and differentially expressed genes that overlap with MRI-guided STIR-positive FSHD muscle biopsies has also been demonstrated in a mouse model with inducible DUX4 expression [80]. High DUX4 and DUX4 target gene expression are also found in blood-derived immortalized FSHD B-lymphoblastoid immune lines [81]. DUX4 expression has also been associated with rheumatoid arthritis and axial spondyloarthritis, characterized by chronic inflammation [82]. Further studies are required to determine why DUX4 produces immune attack in non-transformed tissues but immune evasion in solid cancers.

Surprisingly, DUX4 appears to act as tumor suppressor in colon cancer and synovial sarcoma by multiple mechanisms [14,83]. Understanding the contribution of aberrant DUX4 expression to different tumor types requires additional studies.

DUX4 is also aberrantly re-expressed following infection by all viruses of the Herpesviridae family, which in humans are associated with several diseases, including mononucleosis, encephalitis, varicella and a range of malignancies. Emerging data support a role for DUX4 in the control of viral infection [6,7].

Finally, recurrent translocations giving rise to pro-oncogenic fusion proteins containing DUX4 portions have been described in round-cell sarcoma and acute lymphoblastic leukemia (ALL) [8,9,10,84,85]. In a highly aggressive subgroup of small round cell sarcoma, predominantly affecting children and young adults, the initiating and causative event is a fusion between the high mobility group (HMG) box containing protein Capicua (CIC) and DUX4. This produces a chimeric transcription factor consisting of a large part of CIC, including its DNA-binding HMG box and the DUX4 C-terminus containing its transcriptional activation domain (TAD). While wild-type CIC functions as a transcriptional repressor and a tumor suppressor, CIC-DUX4 is a potent transcriptional activator and a dominant oncogene [86]. In ALL, DUX4 is frequently translocated to the immunoglobulin heavy (IGH) locus. In this case, the rearranged DUX4 always lacks the TAD and is frequently fused at the C-terminus with amino acids encoded by the IGH locus. The resulting protein blocks pro-B cell differentiation inducing leukemia [8,10].

Understanding why early embryos and cancer cells can tolerate DUX4 expression, while DUX4 is highly toxic to muscle cells could provide possible treatments for FSHD. Moreover, therapies developed for FSHD could be used in DUX4-dependent cancers (PMID: 34642317) or vice versa (see Section 10).

## 4. DUX4 Protein Partners

The identification of DUX4 protein partners is clarifying its mechanism of action and could provide novel therapeutic opportunities.

### 4.1. CBP and p300

Choi et al. generated human immortalized muscle cells expressing inducible DUX4 fused with a C-terminal Flag tag. After a minimal induction, DUX4 was immunoprecipitated from nuclear extracts with anti-FLAG antibodies followed by specific elution using FLAG peptides and mass spectrometry to identify eluted proteins. Despite silver stained SDS-PAGE gel displaying many DUX4-Flag associated proteins, the authors reported only two interactors: cAMP-response element binding protein (CBP) and E1A binding protein P300 (p300) [87]. These are protein-lysine acetyltransferases, best known as transcriptional co-activators for several DNA-binding transcription factors [88]. The interaction was confirmed by co-immunoprecipitation and (only for p300) demonstrated to be direct using recombinant protein pulldown. In line with previous results assigning the transcription activation domain to its C-terminus [40,89], the region of interaction with CBP/p300 was mapped to the last 98 amino acids of DUX4. Support for this finding comes also from a recent preprint, which proposes ^416^EYRALL^421^ as the region of DUX4 binding to CBP/p300 [28]. Genome-wide studies indicated that ectopically expressed DUX4 binds to inaccessible chromatin in 60% of cases, while in the other 40% it binds to previously accessible chromatin, probably due to other factors. In both cases, DUX4 binding is associated with a significant increase in histone H3 lysine 27 acetylation (H3K27Ac), which is much more evident for previously inaccessible chromatin. The most strongly upregulated genes are in the proximity of a DUX4 peak. These data suggest that DUX4 acts as a pioneer factor, binding inaccessible chromatin and recruiting CBP/p300 factors to promote local chromatin relaxation and gene activation [28,87]. Intriguingly, induction of DUX4 expression causes a dramatic increase in total H3K27Ac levels suggesting that a large CBP/p300 inactive pool is present and is activated by DUX4 [90]. Accordingly, DUX4 ectopic expression leads to the appearance of a significant number of new H3K27Ac peaks in regions not associated with DUX4 binding. Induced DUX4 also causes an evident decrease in H3K27Ac levels in many genomic regions that are not associated with DUX4. Accordingly, expression profiling shows that most genes downregulated upon DUX4 expression are not direct DUX4 targets, suggesting that their decreased expression is an indirect effect of DUX4 expression, possibly through H3K27Ac rewiring [87]. Similar results have been reported for mouse Dux [1,90], suggesting that these activities might contribute to Dux/DUX4 physiological function in ZGA. Nevertheless, these results have been obtained using systems ectopically expressing Dux/DUX4. Therefore, it would be important to confirm key findings with endogenous Dux/DUX4 in their natural context.

### 4.2. Cytoskeletal and RNA-Binding Proteins

Ansseau et al. used three independent approaches to identify DUX4 interactors [91]. First, a yeast two-hybrid screen of a mouse embryonic cDNA library was conducted using DUX4 as bait. Second, GST-DUX4 purified from bacteria was incubated under low stringency with total protein lysates from HEK293 cells and human myoblasts, followed by tandem mass spectrometry. Third, DUX4 fused to HaloTag was transfected in human muscle cells and total extracts were incubated with HaloLink resin for 96 hours under low stringency, followed by mass spectrometry. The three approaches returned largely non-overlapping datasets. Moreover, most of the putative interactors identified are cytoplasmic proteins, while DUX4 is a nuclear protein. Nevertheless, several interactors were confirmed using either co-immunoprecipitation, pull-down, co-immunofluorescence and/or in situ proximal ligation assay. Among them is the intermediate filament protein desmin, which is mutated in myopathy [92] and several RNA-binding proteins such as FUS or C1QBP that are involved in disease. Notably, FUS aggregates have been reported in FSHD muscle cells [93]. C1QBP is regulated by the glycosaminoglycan hyaluronic acid (HA). Intriguingly, the interaction of DUX4 with C1QBP was independently confirmed recently and a role for HA-mediated signaling in DUX4-induced toxicity was suggested [94].

### 4.3. CDK1

Aberrant DUX4 expression was recently reported to have a specific role in colon cancer [14]. Repression of *DUX4* gene expression was shown to be an important mechanism through which the transcription factors NF-kB and NFE2L3 regulate colon cancer cell proliferation. By performing immunoprecipitation followed by mass spectrometry analysis in HCT116 cells, cyclin-dependent kinase 1 (CDK1) was identified as a DUX4 interactor. The result was validated by co-immunoprecipitation and the interaction was shown to be direct by in vitro pulldown with purified proteins. DUX4 was shown to inhibit CDK1 activity, likely by preventing CDK1 binding to its targets and supporting a tumor suppressor function of DUX4 in colon cancer [14].

## 5. Post-Transcriptional Regulation Mediated by DUX4

Given the centrality of the transcription factor DUX4 in the pathogenesis of FSHD, research has been focusing on the identification of DUX4 transcriptional targets for years. Nevertheless, besides its role as transcriptional activator, increasing evidence indicates that DUX4 has a much broader impact on gene expression. While RNA-seq and Ribo-seq show a high concordance between mRNA abundance and translation status following DUX4 expression [95], quantitative mass spectrometry shows that many genes display discordant transcript and protein levels, with the highest divergence concerning genes downregulated by DUX4 [96]. These results suggest that post-transcriptional modulation by DUX4 occurs primarily at the level of protein stability and are in agreement with previous studies showing that about 30% of transcript-protein pairs display a varying degree of discordance between RNA and protein dynamics [97,98,99] and that the highest discrepancy frequently concerns downregulated transcripts [97]. Future work is needed to establish how DUX4 regulates protein stability. Besides, these results have been obtained using immortalized cells ectopically expressing DUX4 without validation on FSHD muscle cells. Nevertheless, these works underscore the need to validate DUX4-associated disease candidates and biomarkers at protein level and identify novel DUX4-dependent factors and pathways possibly contributing to FSHD pathogenesis [43,96].

Among the factors downregulated by ectopic DUX4 at protein level, with no change or opposite change in transcript abundance, are UPF1, UPF2 and UPF3B (upframeshift protein 1, 2 and 3B, respectively) and XRN1 (5′-3′ Exoribonuclease 1), which are key components of the non-sense mediated decay (NMD) machinery [43,72,96]. NMD is the most conserved cellular mRNA surveillance mechanism, and it is selectively recruited on transcripts harboring premature termination codons or with unusually long 3′UTR, targeting them for degradation [100]. DUX4 causes a strong decrease of UPF1 protein, that is uncoupled from its mRNA level [72]. Conversely, transcripts of other NMD factors, such as *UPF3B* and *SMG7*, are upregulated upon DUX4 expression [72], in line with previous studies showing stabilization of mRNA levels of NMD factor in response to UFP1 depletion [101]. The decrease of UPF1 upon DUX4 expression has been proposed to occur via proteasome-mediated degradation, as MG132-mediated proteasome inhibition restored UPF1 physiological levels in DUX4-expressing myoblasts [72]. Accordingly, several genes involved in the regulation of ubiquitin-proteasome system (UPS) were reported to be upregulated both at transcriptional and protein level following DUX4 induction [40,96]. Nevertheless, previous reports had shown that expression of endogenous DUX4 leads to protein stabilization and lowers protein turnover in FSHD myotubes [102]. This discrepancy might arise from the different model systems and experimental approaches used. Overall, it remains to be established which is the molecular pathway linking DUX4 expression to UPF1 protein decrease, whether the role of DUX4 is direct and if UPF1 downregulation is displayed by FSHD muscle cells.

The alteration of NMD by DUX4 might also take place through the sequestration of EIF4A3 (Eukaryotic initiation factor 4AIII), a component of the exon junction complex, which plays a major role in mRNA surveillance [100]. In DUX4 expressing cells, the bidirectional transcription of *HSATII* repeats results in the formation of intranuclear dsRNA foci, probably required to establish pericentric heterochromatin in the cleavage stage, mirroring what has been described during mouse embryogenesis [103]. It has been shown that DUX4-induced *HSATII* RNAs co-localize with DUX4-induced nuclear dsRNA foci and that these dsRNA foci sequester factors such as ADAR1 and EIF4A3 [54,73].

In agreement with altered NMD, in cells ectopically expressing DUX4 several normally degraded aberrant transcripts were found increased and a DUX4-dose dependent retention of mis- or incompletely-spliced mRNAs was reported [72]. Notably, hundreds of these transcripts appear to be actively translated to produce truncated proteins, including several truncated RNA-binding proteins [43,73]. Hence, it would be important to establish whether aberrant splicing and NMD are present in FSHD patients and if restoration of RNA surveillance to physiological levels ameliorates any of the defects displayed by FSHD muscle cells.

While DUX4 expression inhibits NMD, NMD is an endogenous suppressor of *DUX4*. Notably, *DUX4* mRNA contains a constitutively spliced intron lying within its 3′ UTR downstream of the termination codon, which is a classical substrate of NMD. Importantly, UPF1 knockdown results in an increased fraction of DUX4-positive nuclei in FSHD1 and FSHD2 immortalized myotubes. Therefore, by promoting degradation of UPF1, DUX4 in fact stabilizes its own mRNA [72]. This negative feedback-loop might be a general mechanism controlling DUX4 expression also in healthy cells to avoid unwanted accumulation of a toxic protein.

The post-transcriptional effects of DUX4 on mRNA metabolism also include the increased stability and accumulation of a number of transcripts, among which is mRNA encoding for the transcription factor MYC, with a 2-fold increase in cells ectopically expressing DUX4 and in DUX4-positive FSHD muscle cells [42,73]. Concordantly, ectopic expression of DUX4 is associated with a mild increase in the abundance of proteins of the MYC-mediated apoptotic pathway [73]. On the other hand, this increase is not significant in FSHD cells. Finally, MYC over-expression has been linked to inhibition of NMD [104], thus suggesting a possible contribution to the NMD impairment reported in cells ectopically expressing DUX4. Further work is needed to determine the relevance of *MYC* stabilization in the physiological and pathological activities of endogenous DUX4 (Figure 4).

In addition to the perturbation of RNA metabolism and quality control pathways, a broad consequence of post-transcriptional regulation by DUX4 is the alteration of protein homeostasis, which involves protein mislocalization, aggregation and turnover. In particular, ectopic expression of DUX4 in human myotubes, as well as endogenous DUX4 in FSHD myotubes is associated with altered distribution of ubiquitinated proteins, with increased levels in the cytoplasm and increased number of nuclear aggregates [102]. Similarly, the RNA-binding protein TDP-43, which is normally uniformly distributed in the nucleus, displays a punctate pattern in FSHD myotubes and forms aggregates. While DUX4 also shows a similar punctate staining in FSHD myonuclei, DUX4 and TDP-43 occupy spatially distinct domains, thus suggesting that they do not co-aggregate [102]. Deregulation of protein homeostasis and aggregation of TDP-43 are common denominators of different pathologies besides FSHD, including frontotemporal lobar degeneration (FTD) [105] and amyotrophic lateral sclerosis (ALS) [106,107]. Formation of TDP-43 aggregates in FSHD might be either a loss-of-function mechanism contributing to the disease, as reported for ALS [108], or a consequence of the impairment of proteasome function. Understanding the mechanism underlying DUX4-dependent protein imbalance can be relevant to dissect FSHD pathogenesis, as well as a possible common molecular signature underlying different diseases.

DUX4 expression has been also associated with the alteration of exocytosis. Following DUX4 ectopic expression, muscle cells showed severe Golgi fragmentation, thus suggesting a perturbation of the cellular secretory pathway [96]. However, the direct role of DUX4 on the regulation of this pathway and its relevance for FSHD needs to be further investigated.

## 6. The Role of Mitochondria in FSHD

Skeletal muscle is the largest organ and one of the most energy demanding tissues in the human body [109]. Metabolic impairment might have dramatic consequences for muscle function and could be linked with several pathological conditions [110,111].

Mitochondria are ancestral organelles present in eukaryotic cells that play a pivotal role for the energetic balance and anabolic/catabolic processes of the cells [112]. Indeed, ATP is the key molecule produced by oxidative phosphorylation (OXPHOS) that occurs in mitochondria sustained by tricarboxylic acid cycle through glucose/glycolysis and fatty acids/β-oxidation. Muscle wasting, reduced muscle regeneration and inflammation are only some of the consequences caused by mitochondrial defects [113,114,115].

In FSHD, impaired mitochondrial function [116,117], bioenergetic perturbation [118,119] and increased oxidative stress have been described [68,116,120]. Gene expression profiling and functional evaluations support a decreased mitochondrial complex I and III activity, elevated mitochondrial membrane potential and increased mitochondrial reactive oxygen species (ROS) contributing to oxidative stress generation and apoptosis in FSHD [116,117].

Oxidative phosphorylation, oxidative stress and hypoxia signaling are intimately linked (reviewed in Nat Rev Mol Cell Biol 21, 268–283 (2020). Intriguingly, meta-analysis shows that HIF1-α signaling is perturbed in FSHD and upon DUX4 ectopic overexpression. HIF1-α is the master regulator of hypoxia signaling triggering the transcription of hypoxic genes, including vascular endothelial growth factor. This provides a possible explanation for both increased oxidative stress sensitivity and retinal vasculature abnormalities observed in FSHD [117,121]. Pharmacological inhibition of HIF1-α protein synthesis attenuates cell death caused by ectopic DUX4 overexpression, opening a possible novel therapeutic approach for FSHD [122].

ATP formation is possible thanks to the exchange with free ADP (present in the mitochondrial matrix) mediated by adenine nucleotide translocator (ANT) proteins [123]. ANT1 is the ADP/ATP carrier isoform present primarily in heart and skeletal muscle [124]. Recently, ANT1 was shown also to promote mitophagy independently of its nucleotide translocase catalytic activity [125]. Analogously to *DUX4*, the *ANT1* locus is located at 4q35 and has been shown to be upregulated in FSHD [126,127], especially in the early stages of the disease [128], although with contrasting results [51,129,130]. Molecularly, zinc-finger protein 555 (ZNF555) binding to a *D4Z4* 4qA located enhancer was shown to play a critical role in regulating *ANT1* promoter activity, particularly in FSHD [131]. Intriguingly, transgenic mice with a two-fold increased ANT1 expression progressively develop muscle wasting [132], suggesting that aberrant ANT1 expression could contribute to FSHD variable penetrance.

Mitochondrial accumulation, mitochondrial enlargement and anomalous mitochondrial cristae organization have been reported in FSHD patients [133]. Intriguingly, peroxisome proliferator-activated receptor gamma coactivator 1-alpha (PGC1α), the master regulator of mitochondrial biogenesis, is significantly decreased in FSHD myoblasts compared to healthy controls [133]. *PGC1α* knock-down causes an atrophic phenotype in control myotubes confirming a key role in muscle cell differentiation. A downstream effect of PGC1α reduction is the downregulation of the estrogen-related receptor alpha (ERRα), an orphan nuclear receptor involved in the regulation of specific genes driving mitochondrial biogenesis and a PGC1α cofactor. Treatment with ERRα agonists rescues the hypotrophic phenotype of FSHD myotubes in a PGC1α-independent manner. Hence, reduced PGC1α/ERRα may contribute to the perturbed myogenic differentiation of FSHD muscle cells [133]. However, a direct regulation of PGC1α/ERRα by DUX4 is still missing and if the downregulation of PGC1α is in response to DUX4 expression or mediated by other DUX4-dysregulated players, it needs to be elucidated.

Muscle biopsies from FSHD patients display lipid peroxidation. Moreover, FSHD myoblasts and myofibers from an animal model of FSHD display reduced plasma membrane repair ability [116]. Intriguingly, this defect can be partially rescued by *DUX4* knockdown or by antioxidant treatment [134]. However, further analyses are needed to clarify whether the impaired mitochondrial bioenergetic function is the only source of increased oxidative stress or if other players are involved in the oxidative stress triggering systemic inflammatory response in FSHD.

## 7. Myogenic Pathways Regulated by DUX4

Gene expression profiling indicates a partial block in the normal muscle differentiation program in FSHD and an alteration of several direct targets of the myogenic transcription factor MyoD [135,136,137]. Intriguingly, ectopic DUX4 expression leads to a rapid MyoD downregulation [67,70].

Skeletal muscle regeneration is a finely orchestrated process occurring in adult muscle after degeneration following an injury or in pathological conditions, such as muscular dystrophies. Regeneration is mainly driven by satellite cells, which are quiescent muscle stem cells localized under the basal lamina in resting condition, while activated upon muscle damage [138]. Activated satellite cells proliferate, differentiate and fuse into multinucleated cells in order to restore damaged muscle fibers [139]. 

One of the main factors essential for satellite cell biology is paired box 7 (PAX7) [140]. PAX7 is a transcription factor present in quiescent satellite cells and that progressively reduces its expression during the differentiation. The DNA-binding domain of DUX4 displays a certain degree of amino-acid sequence homology to the PAX7 homeodomain [67], suggesting a possible competition for DNA binding sites in satellite cells that might affect muscle repair capacity in FSHD [45]. In C2C12 muscle cells, Pax7 expression is able to rescue the downregulation of MyoD and counteracts the cytotoxic effect of DUX4 [67]. Intriguingly, suppression of a biomarker consisting of 311 up- and 290 downregulated direct and indirect PAX7 target genes has been associated with FSHD onset and progression [45,141,142]. However, the hypothesis of a DUX4-PAX7 competition requires the presence of the two proteins at the same time and in the same cell. Instead, no co-expression of PAX7 with DUX4 has been shown, thus suggesting a temporal non-overlap between the two proteins [143]. This evidence is in accordance with the myocytes formation ability of cultured satellite cells from FSHD patients and with the expression of DUX4 mainly restricted to FSHD myotubes [4,42] in which PAX7 is usually not expressed.

In addition to MYOD and PAX7, the receptor tyrosine kinase rearranged during transfection (RET) has been recently reported to be essential for myogenesis and downstream of DUX4 [144]. Genetic or pharmacological RET inhibition rescues DUX4-associated differentiation defects, thereby opening a new potential therapeutic strategy for FSHD [144].

The progression from quiescent stem cell to mature postmitotic myofiber is regulated by extracellular signals. Indeed, this process is sustained by a heterogeneous population of cells, including non-myogenic cells, such as immune cells and fibroadipogenic progenitors (FAPs). FAPs are mesenchymal stromal cells located in the muscle interstitium, they are abundant in skeletal muscle, but they are developmentally distinct from myogenic progenitors [145]. FAPs are quiescent in healthy skeletal muscle, but they can rapidly enter the cell cycle in response to acute injury to transiently establish an environment that enhances myogenic differentiation, thus suggesting a role during regeneration. Importantly, FAPs induce differentiation of activated myoblasts by secreting IL-6, IGF-1, Wnt1, Wnt3a and Wnt5a and control satellite cells activation and proliferation [146]. In physiological conditions, FAPs are maintained in undifferentiated state by signals from healthy muscle, while in case of acute muscle damage, inflammatory cells stimulate FAPs proliferation and apoptosis, to allow successful muscle healing and to re-establish the correct basal level of FAPs, in equilibrium with satellite cells. Conversely, upon chronic muscle damage, such as in muscular dystrophies or during aging, where continuous cycles of necrosis and regeneration occur, the balance is shifted toward terminal differentiation of FAPs, and as a result, muscle is often replaced by a mix of fibrous tissue and white adipocytes, in a process termed fibro-fatty degeneration [146]. In this context, there is an increase of PDGFRα (platelet-derived growth factor-α) positive cells, which is a marker of FAPs, that contributes to muscle atrophy.

Very recently, FAPs have been involved in the pathological process caused by DUX4. Bosnakovski and colleagues have shown that in iDUX4pA-HSA mice, in which DUX4 expression is inducible in muscle upon doxycycline administration, there is a remarkable expansion of the fibroadipogenic progenitor compartment associated to fibrosis [147]. The transcriptional profiling of FAPs isolated by iDUX4pA-HSA mice after 10-days or 6-months of doxycycline administration revealed very limited overlap between the two time points, thus indicating an evolving phenotype of FAPs during chronic dystrophy. In particular, the unbalance between pro-inflammatory and anti-inflammatory cells might be responsible of FAPs differentiation. These changes in the FAPs compartment explain why they accumulate instead of returning to the basal level, as occurs in a physiological context after acute injury. 

Although some DEGs identified in FAPs are significantly enriched also in human FSHD biopsies, DUX4 target genes are not elevated in FAPs. This is a critical point that suggests an indirect role of DUX4 in FAPs alteration. FAPs expansion upon doxycycline administration is peculiar of female mice, where the intramuscular accumulation of FAPs was detected in response to DUX4 in a dose- and time-dependent manner. The situation is different in males, showing extremely low levels of *DUX4* mRNA and high levels of DUX4 targets already in absence of doxycycline, which is accompanied by a large increase of PDGFRα+ cells. However, this increase is not further appreciable upon DUX4 induction. These data suggest that chronic low levels of DUX4 might predispose muscle to a profibrotic state. It is tempting to speculate that transient DUX4 expression might induce a long-term alteration of FAPs, which causes a reprogramming of FAPs that is maintained also when *DUX4* is no longer expressed. Indeed, the alteration of profibrotic/adipogenic physiological state associated with DUX4-induction was comparable to that caused by glycerol, which is characterized by an increased number of PDGFRα+ cells at the 10 day time point, rather than cardiotoxin-induced myofiber death, that did not show a significantly different number of FAPs infiltrates after 10 days of injury [80].

The sustained increase of FAPs upon DUX4 induction supports a long-term alteration of the fibroadipogenic state. On the other hand, the mechanism through which DUX4 regulates FAPs needs to be further elucidated, considering that DUX4 expression was not detected in interstitial cells and the induction of DUX4 leads to recruitment of inflammatory cells that might themselves initiate the fibrotic program. To better elucidate the role of FAPs in FSHD pathogenesis, it would be relevant to investigate whether the changes in FAPs compartment observed in iDUX4pA-HSA mice are a feature of FSHD patient muscle biopsies and whether, differently from the mouse model, endogenous DUX4 and its targets are elevated in human FAPs.

## 8. Animal Models of FSHD

Animal models are essential in biomedical research, providing a unique platform to recapitulate human diseases and to test therapeutic approaches that precede clinical trials. However, the genetic and epigenetic complexity of FSHD makes it challenging to generate an animal model that recapitulates every aspect of the human disease. One of the most significant challenges is that the *D4Z4* macrosatellite and the *DUX4* gene are specific to Old World primates [17,18]. Moreover, the proper chromosomal context of the FSHD locus is crucial in the disease [46,63]. Unfortunately, murine Dux only shares partial sequence and functional homology with DUX4 [148]. For these reasons, only some aspects of FSHD can be modeled in animals and by using complementary strategies. For a broad description of the different FSHD animal models that have been generated, we refer the reader to excellent reviews that have been published recently [149,150]. Here, we will focus on recently developed mouse models currently used to understand FSHD pathology and test possible therapeutic approaches. These models have been established according to two general strategies. On one end, transgenic animals exogenously expressing human DUX4 removed from his natural context and upon the control of artificial inducible promoters have been generated. The alternative approach is based on the engraftment of muscle tissue or cultured cells derived from FSHD patients into mice. Both approaches have advantages and drawbacks that have been extensively discussed in [150].

Since DUX4 is extremely toxic, tightly controlled DUX4 expression is required to obtain viable animals [151]. Based on this, three independent transgenic models have been generated expressing DUX4 in an inducible and muscle-specific manner [80,147,152,153,154]. The models differ for the transgenic construct, the site of genomic integration and the induction mechanism, which reflect on differences in disease onset and rate of progression [147,152,153]. Nevertheless, they consistently show DUX4 dose-dependent myopathy characterized by reductions in muscle size and force-generation capacity. Histological analysis shows muscle fiber size heterogeneity, fiber necrosis and central nucleation, mononuclear cell infiltrate, and fibrosis.

In addition to the above changes, one model was characterized by a significant increase in the number of FAPs and a decrease in the number of endothelial cells. Despite these cells do not express transgenic DUX4, they are affected soon after induction of DUX4 expression in muscle [80,147]. Future studies are required to investigate the direct or indirect effects of DUX4 on FAPs and vascular endothelial cells and their possible contribution to the human disease.

A key advantage of inducible models is that DUX4 expression is conditional and titratable, thus the time of onset and the severity of phenotypes can be both controlled to mimic different aspects of the human disease. For example, chronic, low-level, mosaic expression of DUX4 mimics what has been reported in FSHD and can be used to provide a molecular understanding of disease onset and progression. Instead, moderate expression of DUX4 leading to faster progressing disease can be used for therapeutic studies.

Notwithstanding the similarities to the human disease, the transgenic models do not show certain hallmarks of FSHD, like for example the pattern of differentially affected muscles. This could be due in part to transgenic DUX4 expression not recapitulating the one present in FSHD. Moreover, DUX4 is known to regulate very different gene sets in human and mouse [2,47]. Accordingly, while an overlap between differentially expressed genes in FSHD and two of the above models has been reported, the mean overlap is only ~10% or less [80,154]. However, the genes in common between the two species might be involved in important aspects of the pathology.

Intriguingly, sex-specific differences were reported in the models with only males showing a tendency to display phenotypes already in the uninduced condition and females more affected upon DUX4 induction. While these features are not in line with the human disease, they need to be considered when performing phenotypic assays with these models.

In summary, these models are readily scalable and highly reproducible, allowing flexible and tunable development of myopathic features to investigate pathogenic mechanisms downstream of DUX4 or for preclinical testing of therapeutic approaches targeting DUX4 mRNA or protein.

The major limitation of transgenic FSHD models is that human DUX4 is expressed in mouse cells using artificial promoters. Hence, the genetic and epigenetic features of FSHD are lacking in the above models. To address these limitations, alternative models have been developed by engrafting human muscle tissue or muscle cells into the mouse hindlimb to allow for studies of endogenous human gene expression in a living tissue [155,156,157]. One advantage of the transplantation model is that it can be generated using cultured muscle cells to bypass the need of fresh biopsies from a high number of FSHD patients. In this setting, human muscle precursor cells from FSHD patients or control relatives are engrafted into the tibialis anterior (TA) of immunodeficient mice, which have been previously irradiated (to block the activity of the host muscle stem cells) and treated with a muscle damage agent to create a niche for new human-derived muscle to develop. This approach allows robust engraftment, and the genetic and epigenetic profiles of the xenografts reproduce those of FSHD patients, including the expression of key biomarkers [156,158]. Notably, since immortalized FSHD muscle cells can be used for transplantation [144], an almost infinite number of genetically identical grafts can be generated for studies of both FSHD pathophysiology and drug screening.

One obvious advantage over murine transgenic models is that human-to-mouse muscle xenografts are comprised almost exclusively of human tissue and thus provide the best source of mature human muscle, outside of the clinic, to study the specificity and efficacy of drugs designed to treat FSHD. In particular, xenograft models of FSHD are the only setting in which therapeutic approaches targeting the FSHD locus and *DUX4* epigenetic or transcriptional regulation [157,159,160,161,162,163,164,165] can be tested in vivo.

Although powerful, these models display several limitations. For example, they do not allow many functional tests. Thus, the characterization is often limited to histological analysis and gene expression studies. Moreover, since xenograft models require the use of immunodeficient mice, the contributions of the immune system to the pathology cannot be investigated. Therefore, transgenic and xenograft models of FSHD should be seen as complementary systems.

The current array of FSHD animal models offers a significant improvement over previously available models. Nevertheless, future work is required to develop more physiologically relevant models. The use of humanized mouse technology or of species more closely related to humans could allow to reach the next step in in vivo FSHD modeling.

## 9. Therapeutic Approaches

While the genetic defect responsible for FSHD has been known for more than 25 years, no disease-modifying treatment has yet become available. Several clinical trials have been undertaken but, unfortunately, they have produced mixed results [150,157,166,167,168,169,170,171,172,173]. Currently, the only option for FSHD patients is physical therapy. Depending on the individual’s signs, symptoms and functional abilities, the plan may include recommendations about appropriate activities, exercises, modalities for pain relief, guidance for management of fatigue, orthotics/braces, assistive devices, and environmental modifications [174].

Here we will summarize the different approaches currently under study but, for a thorough discussion about the evolution of FSHD clinical trials and the challenges on the road to clinical translation, we refer the reader to recent informative reviews [174,175,176,177].

In recent years, the therapeutic focus has narrowed toward approaches aiming at targeting *DUX4* expression, DUX4 protein or its damaging effects on skeletal muscle.

*DUX4* repression can, theoretically, be reached by reinstating epigenetic silencing at the FSHD locus, by blocking *DUX4* transcription or by degrading *DUX4* mRNA.

There are two main strategies being developed to reconstitute the epigenetic silencing of the FSHD locus. Based on the fact that *SMCHD1* mutations are associated with *D4Z4* DNA hypomethylation leading to FSHD2 and to increased severity of FSHD1, *SMCHD1* gene correction with CRISPR-Cas9 was performed in primary and immortalized muscle cells from one FSHD patient. This leads to repression of DUX4 and its targets. Surprisingly, DNA hypomethylation was not corrected [162]. Given that SMCHD1 has been reported to collaborate with the Polycomb Repressive Complex 1 (PRC1) in X-chromosome inactivation [178,179] and that PRC1 is involved in silencing of the FSHD locus [46,180], it is tempting to speculate that *SMCHD1* gene correction acts together with PRC1 to reinstate epigenetic silencing of the FSHD locus.

Another strategy to increase epigenetic silencing at the FSHD locus is to use an enzymatically inactive “dead” Cas9 (dCas9) fused to an effector domain from an epigenetic repressor (CRISPRi) [181]. Using this approach, repression of *DUX4* and its targets has been achieved in cell lines and in FSHD muscle cells [160,163,165].

While the above results are promising, the repression of *DUX4* expression is partial and no phenotypic improvement has been reported. Moreover, a key issue of these approaches is efficient delivery to skeletal muscle in vivo [175]. Indeed, attempts to silence *DUX4* expression by CRISPRi in an animal model were associated to modest gene silencing results and no amelioration of disease symptoms was reported [163].

Several screenings have been performed to identify drugs able to inhibit *DUX4* transcription [95,157,182,183]. Of these, the most promising is the identification of the p38 mitogen-activated protein kinases inhibitor losmapimod. In vitro and in vivo data indicate that p38-α (MAPK14) and p38-β (MAPK11), through an unknown mechanism, are required for *DUX4* expression and treatment with losmapimod improves disease phenotypes [157,183]. Based on this and since losmapimod has a good safety profile, a phase 2b clinical trial sponsored by Fulcrum Therapeutics has been performed (NCT04003974). While the primary endpoint, change in DUX4-driven gene expression, was not met, statistically significant and clinically relevant benefit was recently reported (https://ir.fulcrumtx.com/news-releases/news-release-details/fulcrum-therapeutics-announces-results-redux4-trial-losmapimod) (accessed on 22 November 2021).

A number of academic groups and companies are developing ways to degrade *DUX4* mRNA using antisense oligonucleotides, siRNAs, miRNAs, compounds or CRISPR approaches. The most frequent readout is a decrease in levels of DUX4 and its targets, and sometime improvement in phenotypes, in tissue culture. In some cases, promising results have also been reported by in vivo treatments. Ways to improve skeletal muscle-specific delivery are currently being investigated in order to increase safety and efficacy, as extensively discussed in recent reviews [175,177].

Several groups have reported molecules able to induce degradation of DUX4 protein or to interfere with its activity, including the inhibitor of hyaluronic acid biosynthesis 4-methylumbelliferone, hypoxia signaling inhibitors, the estrogen receptor agonist 17β-estradiol, the inhibitor of the receptor tyrosine kinase Ret Sunitinib and DNA aptamers [24,94,122,144,170]. While the exact molecular mechanism underlying the effects is not always clear, these studies reported significant improvement in a number of parameters. Curiously, several of these treatments do not affect the expression of DUX4 targets, which are considered intimately linked in FSHD [40,42,87,184,185], raising concern regarding some of the readout currently used in the field.

Small molecule screenings identified a high number of compounds reducing oxidative stress as inhibitors of DUX4-induced toxicity [44,186]. Intriguingly, mitochondria-targeted antioxidants appear to rescue myogenic defects of FSHD muscle cells more effectively than conventional antioxidants [117]. However, a direct effect of these compounds on DUX4 or its downstream effectors needs to be clarified.

Recently, a new p300 inhibitor, iP300w, has been tested as a possible therapeutic strategy for FSHD. The compound inhibits DUX4-induced transcription and DUX4-associated cytotoxicity in cells and mice ectopically expressing DUX4. Importantly, iP300w can also inhibit DUX4 transcriptional targets in muscle cells of FSHD patients [90]. Notably, iP300w also caused significant global alterations in DUX4 negative cells. Hence, long-term iP300w treatment would probably be associated with unwanted side effects. Moreover, almost half of direct DUX4 targets are located in accessible chromatin, which might not be affected by a p300 inhibitor. Nevertheless, this is a crucial proof of principle justifying the search of selective inhibitors of DUX4 interaction with p300 [90].

Given that DUX4 affects multiple biological processes, it is surprising that targeting a single pathway could be of therapeutic efficacy. It is tempting to speculate that several of the above molecules converge on the same molecular signaling contributing to FSHD.

While it is currently unknown how much DUX4 repression/inhibition is required to achieve a clinically relevant result, there are indications that even a small reduction could be sufficient to obtain a therapeutic benefit. Moreover, many of the approaches under development are not alternative but could also be combined in a synergistic action opening new and exciting perspectives for the possibility to develop a safe and effective therapy for FSHD patients.

## 10. Conclusions

In this review we summarized the key cellular and molecular aspects of FSHD, one of the most common muscular dystrophies afflicting children and young adults, regardless of gender. FSHD is characterized by a slow muscle degeneration that eventually forces patients to a wheelchair. In contrast to the classic Mendelian diseases, FSHD is not caused by a mutation in a gene sequence. FSHD represents a unique case for a muscular dystrophy since the expression of DUX4, normally restricted at the embryonic development, is aberrantly reactivated at the wrong time and in the wrong place.

Due to its pivotal role in the onset and the progression of the pathology, DUX4 has been studied in both cellular and animal models of FSHD. However, despite the relevant knowledge acquired in recent years, several questions remain unanswered (Figure 5).

Despite several clinical trials, there continues to be no cure or therapeutic option available to FSHD patients. However, the consensus that ectopic DUX4 expression in skeletal muscle is the root cause of FSHD pathophysiology has opened the possibility of targeted therapies.

Identifying DUX4 interactors is crucial to understand possible DUX4 regulators and the pathways involved in DUX4 toxicity, which could help designing small molecule inhibitors. Importantly, it has been shown that the ability of DUX4 to activate its direct transcriptional targets is required for DUX4-induced muscle toxicity. Accordingly, DUX4 targets account for the majority of gene expression alterations in FSHD skeletal muscle. Thus, blocking the ability of DUX4 to activate its transcriptional targets is expected to have strong therapeutic relevance.

The involvement of DUX4 in post-transcriptional regulation of gene expression in FSHD emerged only recently. The way this regulation occurs as well as the direct implication of DUX4 in all the described mechanisms still remain to be fully elucidated.

A key issue is that most of the studies conducted thus far have been carried out in cellular systems upon ectopic expression of DUX4, which are useful to identify and dissect molecular pathways, but can differ from FSHD muscle primary cells or biopsies, where endogenous DUX4 has a dynamic and sporadic expression pattern, mainly restricted to differentiating cells.

Overall, further studies are required to dissect the complex regulatory landscape mediated by DUX4 and link it to the aspects underlying FSHD disease to develop effective treatments.

## Figures and Tables

**Figure 1 cells-10-03322-f001:**
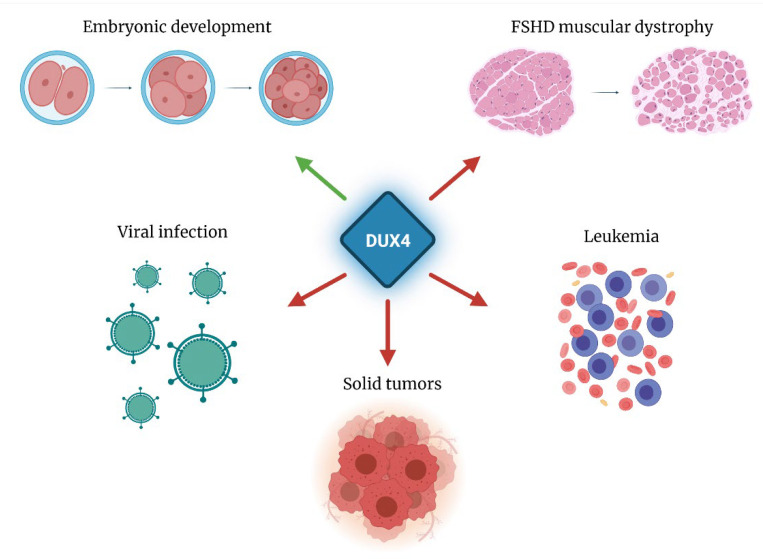
Physiological (green arrow) and pathological (red arrows) roles of DUX4. DUX4 is physiologically expressed during early embryogenesis and subsequently silenced in most somatic tissues. DUX4 can be re-expressed as direct consequence of Herpesviridae infection. DUX4 pathological gain of expression is associated with FSHD muscular dystrophy. Several forms of neoplasms display aberrant DUX4 expression or activity.

**Figure 2 cells-10-03322-f002:**
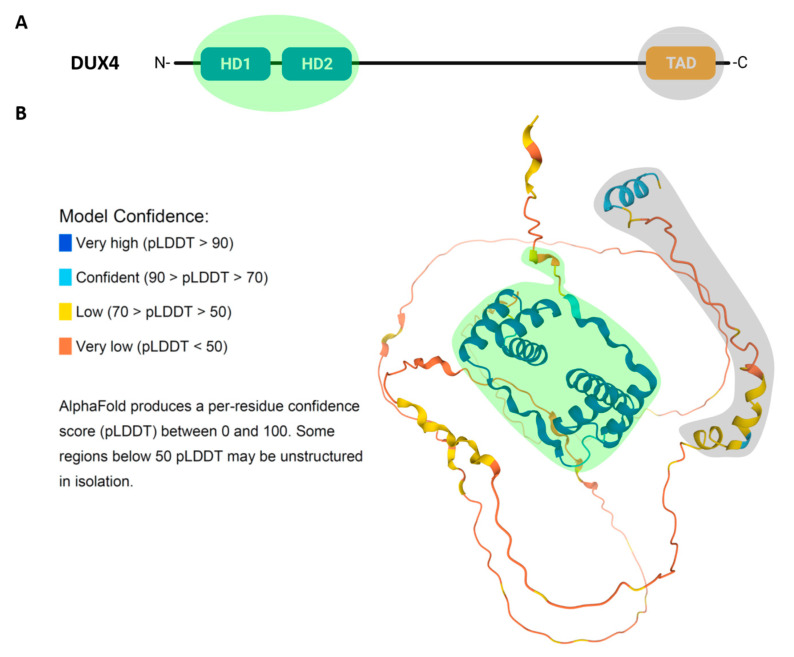
(**A**) Schematic representation of DUX4 containing two homeodomains (HD1 and HD2) at the N-terminus (green shading) and the transactivation domain (TAD) at the C-terminus (grey shading). (**B**) Prediction of DUX4 3D structure using AlphaFold. DUX4 DNA binding domain (green shading) and the C-terminal region containing the DUX4 transcription activation domain (grey shading) have a well-defined tertiary structure, while the rest of the protein is predicted to be mainly disordered (unstructured).

**Figure 3 cells-10-03322-f003:**
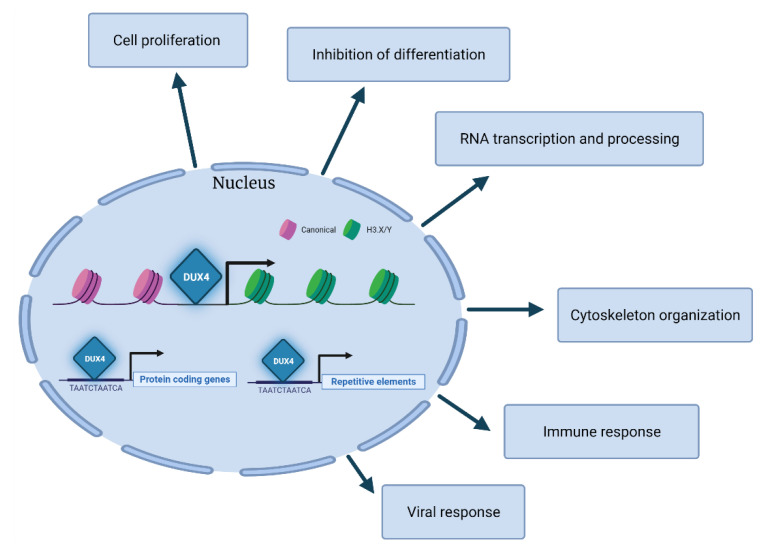
DUX4 transcriptional pathways. Main pathways modulated by DUX4 across different studies highlight several processes, including proliferation, inhibition of cell differentiation, RNA transcription and processing, cytoskeleton organization, immune response and viral response.

**Figure 4 cells-10-03322-f004:**
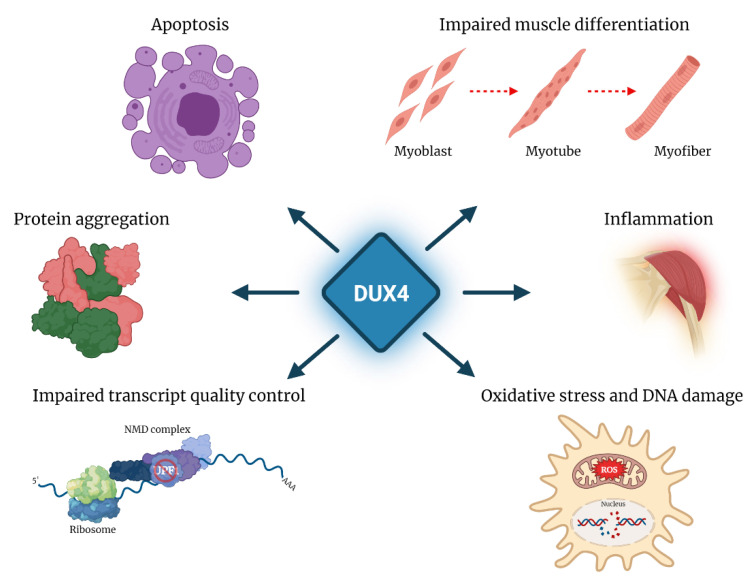
Pathological outcomes of DUX4 aberrant expression. DUX4 expression in FSHD is associated with the activation of toxic pathways, such as inhibition of myogenic differentiation, inflammation, oxidative stress and DNA damage, impaired transcript quality control, protein aggregation and apoptosis.

**Figure 5 cells-10-03322-f005:**
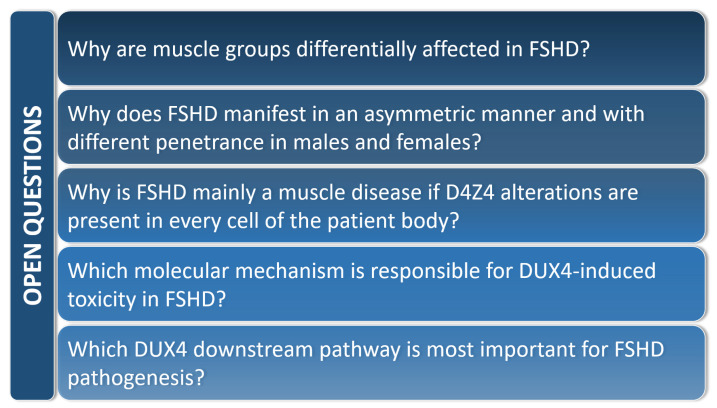
Relevant open questions in the FSHD field.

## Data Availability

Not applicable.

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
