# Peer review of "DUX4 Role in Normal Physiology and in FSHD Muscular Dystrophy"

_cells, 2021, doi:10.3390/cells10123322_

Round 1

Reviewer 1 Report

A well-written review. some minor recommendation

  1. on page 2, line 50: "Liquid" cancer is not how we normally describe hematological tumors.  simply change "solid and liquid cancers" to "neoplasms"?
  2.  The reviews is a little too long. I think you can consider simplifying the paragraphs on page 12 as there is not enough evidence to correlate the mitochondrial dysfunction in FSHD with DUX4.  

Reviewer 2 Report

Dr. Emanuele Mocciaro and colleagues described the roles and functions of Dux4 in transcription of zygotic genes shortly after/during mid-blastula transition in which zygotic genome activation is triggered. Abnormal activation of DUX4 after embryogenesis causes/is induced in several diseases, including tumorigenesis and FSHD muscular dystrophy, is also addressed.  The 5 figures provided here are well organized and very helpful for readers to understand the content of this review.  The topics selected and their description/explanation are both very professional, so this review is almost ready for its publication in the Cells journal.  However, a few writing typo errors are found and they should be corrected before publication.  Here is the list of these typo errors:

  1. The language is good, but more editing can make it more legible for the readers.
  2. In 2.1, please indicate whether Dux4 binds to DNA as a monomer or dimer, it help clear the mind of a new researcher in this field.
  3. Page 6: The pathology and consequences of FSHD should be described in more detail.
  4. Page 4, line 105: pick should be peak.
  5. Page 4, line 140: will detectable better than appreciable?
  6. Page 4, line 146 T-cell receptor ?-chain.
  7. Page7, line 235: please define STIR-POSITIVE.
  8. Page 7, paragraph 2 and 3 should be further modified. Does Dux4 induce immune evasion in colon cancer?
  9. Page 10, paragraph 1: Does the reduced NMD proteins increase non-sense mRNA level and induce phenotypes?
  10. Page 10, last paragraph: Will Myc induction affect the tumor suppressor role of Dux4?
  11. Page 11, paragraph: Is there a proposed mechanism by which Dux4 affect protein homeostasis?
  12. Line 464: more can be changed to most.
  13. Lines 522-524: please re-phrase the sentence.
  14. Line 531: Some examples are needed.
  15. Line 667: differently should be differentially
  16. Line 934: The question “Which is the molecular mechanism responsible for DUX4 toxicity?” can be changed to “Which molecular mechanism is responsible for DUX4 toxicity?”

Reviewer 3 Report

The paper from Mocciaro et al. is interesting and up to date with the latest publications and studies in progress. This review has the particularity of treating in details the physiological roles of DUX4, which is scarcely developed in other review.  The paper focus on FSHD but still touches on other interesting pathological aspects related to DUX4 (cancers etc.) which surprisingly sometimes present contradictory pathological processes (especially at the inflammatory level).   Please insert a summary diagram of genetics in FSHD 1 and 2 for readers that are not familiar with myopathy.    Many biomarkers have been put forward in the context of the FSHD but some are maybe missing (H3K9m3, DBET, etc.).   The general organization is synthetic and well organized. Citations/references need to be better formatted. Specific sections:    "2.1 The DUX4 protein structure and DNA binding"   The presentation of the 3D structure of DUX4 is a good idea (Figure 2). However, I think it would be interesting to add visual annotations, including alpha propellers etc.     "2.3 DUX4 in somatic tissues"   Correction to be made on "@-chain" (line 146)   "3. Transcriptional effects"  Figure 3: I would have added cell death in this diagram also since apoptosis is also presented as a physiological process (described in 2.3).   "4. Pathological role of DUX4" I think there is an extra spacing between "general" and "reduction" (line 208).   "Generally more than 8 units" (line 212): There is no reference given here for this figure which does not represent in my opinion the threshold healthy patient / sick patient usually stated. Moreover, this is not consistent with the value given in the following line. A clearer separation such as sub-parts in the manuscript between the FSHD part and the Cancers, virus etc part is maybe clearer. "5. DUX4 protein partners"   Apart from the rather well developed sub-part 5.1, I find the following part too synthtic especially the sub-part "5.2 Cytoskeletal and RNA-binding proteins" for which I expected and I would have liked a more in-depth development. «8. Myogenic pathways regulated by DUX4»   Very good initial presentation followed by a very interesting and very developed passage on FAPs (a link of which is also made in part 9) but I would have liked something more diversified in this part. I sometimes have the impression that the authors focus  on specific aspects when I would sometimes like to have a greater vision of things as expected by the titles.   "9. Animal models of FSHD"   Pro & Cons very well highlighted for both proposed alternatives.   "10. Therapeutic Approaches"   I was initially surprised that some upstream sub-parts also contained some descriptions of therapeutic studies (P300 inhibitor, antioxidants, RER inhibition), this could be understandable given that these studies were directly related to the developed sub-part. But all of these studies have been stated again here (except perhaps antioxidants).   "Conclusion"   Although the whole paper is written around the FSHD and given the title on DUX4, I would have appreciated a small concluding sentence at the beginning to summarize the physiological role of DUX4 and the whole part relating to other pathologies (cancers ...) before developing everything on FSHD. 

Reviewer 4 Report

The work is a well-written, comprehensive and informative review on DUX4, its functions and its less discussed roles in pathologies other than FSHD, before latterly focussing on FSHD. The supporting diagrams are elegant and summarise key findings, but are slightly simplistic and could convey more information. The ‘Conclusion’ and ‘Open Questions’ sections focus only on FSHD and should be expanded to reflect the wider remit of the review that mentions DUX4 in normal physiology and other pathologies.

The only major point is that the names in citations in the text and bibliography are only partial (initials) in places.

My comments for consideration are minor.

  1. Very occasional minor changes in syntax required (e.g. line 525, 554, 596).
  2. Figure 1 – could rearrange/colour code to distinguish ‘normal’ functions of DUX4 from its roles in pathology.
  3. Section 2.1 - would be helpful to mention relevant species here and that DUX4 is restricted to old world primates, rather than later in the piece. Should mention DUX4c here too.
  4. Figure 2 – Text of key is very small and hard to read. Could add a linear schematic of the DUX4 protein to show functional domains. Comparisons with other prominent Dux genes could be useful, but not essential.
  5. Line 105 – ‘peak’ rather than ‘pick’.
  6. Section 2.3 – could also mention potential roles associated with the immune status of these tissues too (testis – immune privileged, thymus – immune system maturation).
  7. Line 146 – Greek symbol corrupted.
  8. Line 170 – would be clear about the species that the data is derived from.
  9. Line 208 – would reword term ‘medical consumption’.
  10. Section 4 – need to mention the DUX4 polyA signal and permissive haplotypes, maybe around line 219.
  11. Line 249 – role of DUX4 in the control of viral infection is relatively recent and could be expanded upon here.
  12. Line 265 – many different hybrid proteins made, not a uniform protein.
  13. Line 315 – the iP300w therapeutic point could be moved to therapeutics section, or at least mentioned again there.
  14. Section 7 – would include ‘Free Radic, Biol Med. 2012 Sep 1;53(5):1068-79. doi: 10.1016/j.freeradbiomed.2012.06.041. PMID: 22796148’.
  15. Section 7 - although putative mechanisms of oxidative stress generation and mitochondrial dysfunction are comprehensively outlined, there is an inevitable relation between OXPHOS, oxidative stress and hypoxia signalling. Recent work has identified HIF1a dysregulation in FSHD patients (org/10.1098/rsif.2014.0797) and in DUX4 overexpression models (doi.org/10.1101/2021.09.08.459509). Importantly, HIF1a signalling directly or indirectly convers DUX4 toxicity, which may be a therapeutic entry point (doi.org/10.1126/scitranslmed.aay0271). Since p300/CBP, which are mentioned in the review, act together with HIF1a, I think this section would benefit from mention of this pathomechanism.
  16. Section 8 - Title of ‘Myogenic pathways regulated by DUX4’ fine for initial part, but not sure that it really encompasses the FAPS section. Separate title? Also no paragraphs in that part on page 14.
  17. Section 8, Line 550 – would clarify that it refers to the homeodomain of PAX7, as PAX7 also binds DNA via its paired domain.
  18. Section 8, Line 553 – would mention that this data is from mouse C2s.
  19. Line 559 onwards – this is modelled myogenesis using human isogenic iPS and ES cells. Indeed as Haynes et al 2017 themselves state in the abstract ‘While these studies examine DUX4, PAX3, and PAX7 expression patterns during stem cell myogenesis, they should not be generalized to tissue repair in adult muscle tissue.’ Co-expression in immortalised myoblast lines has been shown (e.g. doi: 10.1038/s41467-017-01200-4). The Rickard et al 2015 paper also finds DUX4 reporter activity in FSHD primary myoblasts. Although more in myotubes, it is still only a small proportion of total cells.
  20. Section 9 – think that the model where mice were made transgenic for a healthy and FSHD human locus are worth a mention. Although not as useful as hoped, they do provide a hybrid approach between non-native DUX4 drivers and human muscle/cell grafting (PLoS Genet. 2013 Apr;9(4):e1003415. doi: 10.1371/journal.pgen.1003415. Epub 2013 Apr 4. PMID: 23593020 and J Cell Sci. 2016 Oct 15;129(20):3816-3831. doi: 10.1242/jcs.180372).
  21. Section 9, line 698 – Moyle et al (Elife. 2016 Nov 14;5:e11405. doi: 10.7554/eLife.11405) grafted immortalised FSHD myoblasts.
  22. Section 10, line 722 – this is the wrong set of citations.
  23. Latter half of the review is text-heavy and would benefit from another figure. Summarising the modes of action of DUX4 would be useful in figure or table form.
  24. Conclusion – Just focussed on FSHD, whereas the abstract and initial parts are about DUX4 and its roles in embryogenesis and various diseases, so should be amended to include the wider content to make the review more cohesive.
  25. Open Questions – similar to conclusions section, focuses exclusively on FSHD.

Reviewer 5 Report

The authors put a long review on DUX4 molecular function and role in human pathologies.

This is a relevant subject for a review. However, in its current form the review is not well structured. The focus on both FSHD and DUX4 is already highly ambitious, as it covers both pathological and clinical aspects as well as molecular aspects of the disease.

The authors included also other, non-FSHD, pathologies, which are scattered in the review without a clear structure.

As a result, reading this review is quite tiresome.

My suggestions is to focus on one subject either DUX4 in development and human pathologies or on DUX4 in FSHD as the title indicates and the conclusion (and leave other pathologies out).

In addition, the manuscript requires editing as it is scattered with spelling/grammar/errors and unclear sentences (only a few examples are depicted here).

  1. 10: “to a key factor for normal physiology and important diseases.”
  2. 17: “not only to further our understanding of”
  3. 61 “and encoded by repeated elements”
  4. 79: “structural studies about how DUX4 binds DNA”

L130-131: “indicating that Dux is 130 important but not strictly essential for early embryo development”, please rephrase. Do you mean DUX4 is not a vital gene? What do you mean “important”?

Line 206: should be van der Maarel

Lines 305-306 “Expression profiling show that gene downregulated upon DUX4 expression belong almost exclusively to the set not associated with DUX4 binding (SH et al., 2016).” is unclear.

Other comments are related to figures and content are examples of unstructured text.

Figure 2: the text font in the figure is not readable. Please indicate HD1 and HD2 in the model, as it is indicated in the text. Please define “disordered” (structure).

Lines 78-93 describe DUX4 structure. It will be clearer if the structures that are discussed in these lines are shown in Fig. 2. In the current presentation is it unclear.

The section of UDX 4 function in T-cells (L139-151) is highly speculative and not convincing. It is unclear how many DUX4 homologs are there in mouse, and how you determine these are DUX4 homolog and not another member of the DUX gene family.

Lines 152-153 are redundant, it is not the right place (after the section on Dux4 regulation of T-cells.

Lines 178-180: “The possibility that DUX4 down-178 stream pathways might persist in the absence of DUX4 provide a possible explanation as to why it is easier to detect DUX4 target genes than DUX4 itself, but raises concern regarding therapeutic options directly targeting DUX4.” is not sufficiently clear. What is the relation for “therapeutic options”?

Line 191-192: “their expression in muscle cells could give rise to a transcriptional rewiring 191 thus contributing to FSHD pathology.” requires more explanation.

The subhead “Pathological role of DUX4” comes after FSHD is mentioned few times. This is not a logical order.

The muscle tissue is not mentioned in section 2.3 “somatic tissues”

The sentence “raising direct and indirect socioeconomic costs due to medical consumption, work productivity loss and a general reduction in health-related quality of life(Blokhuis et al., 2021).” Reads as if it is taken from a grant proposal. How it is related to DUX4?

The sentence “Understanding why early embryos and cancer cells can tolerate DUX4 expression, 268 while DUX4 is highly toxic to muscle cells could provide possible treatments for FSHD. 269 Moreover, therapies developed for FSHD could be used in DUX4-dependent cancers or 270 vice versa.” comes after three paragraphs on DUX4 in cancers. This is confusing.

Figure 3 and 4 are partly redundant.

Figure 5 is not a figure: just text.
